# Identification of Circular RNAs in the Anterior Pituitary in Rats Treated with GnRH

**DOI:** 10.3390/ani11092557

**Published:** 2021-08-31

**Authors:** Hai-Xiang Guo, Bao Yuan, Meng-Ting Su, Yi Zheng, Jin-Yu Zhang, Dong-Xu Han, Hao-Qi Wang, Yi-Jie Huang, Hao Jiang, Jia-Bao Zhang

**Affiliations:** Department of Laboratory Animals, College of Animal Sciences, Jilin University, Changchun 130062, China; guohx20@mails.jlu.edu.cn (H.-X.G.); yuan_bao@jlu.edu.cn (B.Y.); 13704550931@163.com (M.-T.S.); yizheng18@mails.jlu.edu.cn (Y.Z.); 0805zjy@163.com (J.-Y.Z.); handx18@mails.jlu.edu.cn (D.-X.H.); hqwang1997@163.com (H.-Q.W.); huangyj18@mails.jlu.edu.cn (Y.-J.H.)

**Keywords:** rat, anterior pituitary, circular RNAs, GnRH, animal reproduction

## Abstract

**Simple Summary:**

The pituitary gland, an important endocrine organ, can secrete a variety of reproductive hormones under the action of hypothalamus-secreted gonadotropin-releasing hormone. Circular RNAs are a class of RNA molecules with stable covalently closed circular structures. In this study, we performed RNA sequencing of GnRH-treated rats to identify differentially expressed circRNAs in the anterior pituitary. The results revealed 1433 related circRNAs, 14 of which were differentially expressed. We predicted targeted relationships between the differentially expressed circRNAs and FSHb-LHb-associated miRNAs. In all, a total of 14 circRNAs were identified that may act on the secretion and regulation of reproductive hormones in GnRH-treated rats.

**Abstract:**

The pituitary gland, an important endocrine organ, can secrete a variety of reproductive hormones under the action of hypothalamus-secreted gonadotropin-releasing hormone (GnRH) and plays important roles in animal reproduction. Circular RNAs (circRNAs) are a class of RNA molecules with stable covalently closed circular structures. CircRNAs are equipped with miRNA response elements (MREs), which can regulate the expression of target genes by competitively binding miRNAs. However, whether the expression levels of circRNAs in the pituitary gland change under the action of GnRH and whether such changes can further affect the secretion of reproductive hormones are still unclear. In this study, we performed RNA sequencing (RNA-Seq) of GnRH-treated rats to identify differentially expressed circRNAs. The results revealed 1433 related circRNAs, 14 of which were differentially expressed. In addition, we randomly selected five differentially expressed circRNAs and tested their relative expression levels by RT-qPCR, the results of which were consistent with the RNA sequencing results. Finally, we predicted targeted relationships between the differentially expressed circRNAs and FSHb-LHb-associated miRNAs. In all, a total of 14 circRNAs were identified that may act on the secretion and regulation of reproductive hormones in GnRH-treated rats. Our expression profiles of circRNAs in the anterior pituitaries of rats treated with GnRH can provide insights into the roles of circRNAs in mammalian development and reproduction.

## 1. Introduction

Mammalian GnRH is a decapeptide molecule previously known as luteinizing hormone-releasing hormone (LHRH) [1]. It can participate in the regulation of vertebrate reproductive function through the hypothalamic-pituitary-gonadal axis [2]. Studies have shown that this neuropeptide is synthesized and released by hypothalamic neurons and then enters the anterior pituitary mainly through the hypophyseal portal circulation [3,4]. The function of GnRH is to stimulate pituitary synthesis and release luteinizing hormone (LH) and follicle-stimulating hormone (FSH), to regulate the secretory activity of the pituitary, and ultimately to affect the gonads, playing a pivotal role in reproduction [5,6]. GnRH binds to and activates its corresponding receptor (GnRHR) on gonadotropic cells of the pituitary gland, stimulating the synthesis and secretion of gonadotropins by these cells [7,8]. GnRH can participate in the regulation of a variety of reproductive processes in mammals. For example, the addition of GnRH agonists during artificial insemination can induce ovulation and improve embryo survival [9,10]. GnRH regulates the production of luteal progesterone and testosterone in different species [11,12,13,14]. In addition, GnRH agonists can also participate in the reproductive regulation of mammals after pregnancy [15]. GnRH analogs have been researched and developed and are currently used extensively in treatment regimens, such as those for in vitro fertilization and ovulation in women [16,17].

The pituitary gland is an important part of the endocrine center [18]. Various hormones are secreted by the pituitary gland, such as growth hormone (GH), follicle stimulating hormone (FSH), adrenocorticotropic hormone (ACTH), prolactin (PRL), and arginine vasopressin (AVP) [19]. A variety of factors participate in the reproductive process by regulating the secretion of pituitary hormones. Kisspeptins (Kiss) are neuropeptides that can regulate the secretion of FSH and LH from the pituitary gland by acting upstream of GnRH [20,21]. In addition, pituitary hormone release is controlled by PtdIns (3,4,5) P3-dependent signal transduction through GPCRs [22]. The pituitary is also regulated by prostaglandins and n-3 polyunsaturated fatty acids [23]. The pituitary gland is part of the hypothalamic-pituitary-gonadal axis, which regulates reproduction [24]. This axis produces gonadotropins, mainly FSH and LH [25], and plays important physiological roles. Recent studies have shown that the HPG axis regulates the reproductive process in a time-dependent manner, and circadian rhythm disorders can lead to diseases, such as polycystic ovarian syndrome (PCOS) and premature ovarian insufficiency (POI) [26].

Circular RNAs (circRNAs) are newly identified RNAs that, unlike linear RNAs, exhibit stable covalently closed circular structures [27,28]. Recently, thousands of strongly and stably expressed circRNAs have been detected in humans and animals [28], and these circRNAs exhibit tissue-developmental stage-specific expression [29]. Since circRNAs usually do not have poly-A tails, they have greater stability and higher sequence conservation than normal linear RNA molecules (such as microRNAs [miRNAs] and long noncoding RNAs) in mammalian cells [30]. Although the exact functions of circRNAs are unknown, an increasing number of reports have indicated that circRNAs can act as miRNA sponges by binding to miRNAs to regulate gene expression. For example, in lung cancer, CircAGFG1 sponges miR-203 to regulate the expression of the Znf281 gene [31], and in postmenopausal osteoporosis, circRNA-0016624 can regulate Bmp2 expression via miR-98 [32]. In addition, an increasing number of reports have indicated that circRNAs play key roles in a variety of biological processes. In 2019, Han et al. identified circRNAs in the anterior pituitaries of mature and immature rats [33]. However, information on circRNAs in the rat pituitary, particularly after GnRH treatment, is still limited.

In this study, we performed RNA sequencing of the pituitaries of rats treated with GnRH. Then, we analyzed the differentially expressed genes. Gene Ontology (GO) and Kyoto Encyclopedia of Genes and Genomes (KEGG) analyses were performed on the differentially expressed circRNAs. In addition, we randomly selected differentially expressed circRNAs and detected their relative expression levels by RT-qPCR. Finally, based on the results of our previous research [34] identifying many miRNAs that may have targeting relationships with Fshβ and Lhβ, we predicted the differentially expressed circRNAs that may target these miRNAs. Our results provide a powerful resource for broader study on the regulatory functions of circRNAs in rats and contribute to a better understanding of mammalian reproduction and development.

## 2. Materials and Methods

### 2.1. Ethics Statement

The experiments were performed in strict accordance with the guidelines of the Guide for the Care and Use of Laboratory Animals of Jilin University. In addition, all experimental protocols were approved by the Institutional Animal Care and Use Committee of Jilin University (Permit Number: 201809010).

We performed animal experiments in the Laboratory Animal Center of Jilin University, and all experimental protocols were approved by the Institutional Animal Care and Use Committee of Jilin University (Permit Number: 201809010). The animals were anesthetized with carbon dioxide, and finally the animals were euthanized with carbon dioxide.

### 2.2. Collection of Tissue Samples

Eighteen healthy 8-week-old sexually mature male Sprague Dawley (SD) rats were used in this study. Sprague-Dawley rats were provided by the laboratory Animal Center of Jilin University. Twelve of them were combined and sequenced and analyzed. The other six were equally divided into two groups to detect genes and hormones.

We randomly divided twelve rats into four groups, and each group contained three rats. Two of them were the experimental group (L01, L02), and the other two were the control group (L03, L04). The control group was injected subcutaneously with 0.2 mL of normal saline. According to the European Medicines Agency’s summary report on GnRH, the experimental group was injected with 0.2 mL of prepared GnRH solution (containing 0.2 µg of drug) (Appendix A), which was prepared from Gonadorelin for injection (San Sheng, Ningbo, China). The rat pituitary gland was removed and washed in PBS solution. The anterior pituitary lobe was isolated from the pituitary, placed in a centrifuge tube containing TRIzol reagent, and then stored at −80 °C until RNA extraction.

We divided the remaining six rats into a control group and an experimental group. The control group was injected subcutaneously with 0.2 mL of physiological saline, and the ex-perimental group was injected with 0.2 mL of GnRH solution (1 μg/mL) prepared from Gonadorelin for injection (San Sheng, Ningbo, China), two hours later, blood was collected from the tail vein of each rat and immediately stored at 4 °C. The rats were euthanized after the blood was collected. The rat pituitary gland was removed and washed in PBS solution. The anterior pituitary lobe was isolated from the pituitary, placed in a centrifuge tube containing TRIzol reagent, and then stored at −80 °C until RNA extraction.

### 2.3. ELISA

We used a Rat FSH ELISA Kit and a Rat LH ELISA Kit according to the manufacturer’s instructions (Haling Biotech Co., Ltd., Shanghai, China) to measure the FSH levels and LH levels under different experimental conditions.

### 2.4. RNA-Seq

To ensure the quality of the data obtained, we first subjected the RNA to quality control. We used a NanoDrop instrument to detect the purity of the RNA sample, a Qubit 2.0 instrument to detect the concentration of the RNA sample, and an Agilent 2100 instrument to detect the integrity of the RNA sample. Electrophoresis was performed to determine whether the RNA was contaminated with DNA. After the test samples were qualified, the samples were pooled, and a library was constructed. rRNA was removed from the samples using an Epicenter Ribo-ZeroTM Kit, and PCR amplification was performed to obtain cDNA libraries according to the company’s recommended method. The effective concentration of the library was accurately quantified by RT-qPCR. After the library was qualified, the samples were sequenced using an Illumina HiSeq platform (NEB, Ipswich, MA, USA).

### 2.5. CircRNA Identification

The CIRI [35] software compares with the reference gene sequence to generate an SAM file, and analyzes the CIGAR value in the SAM file, scanning PCC signals (paired chiastic clipping signals) from the SAM file. The characteristic CIGAR value in the junction reads is xS/HyM or xMyS/H, where x and y represent the number of bases, M is mapping, S is soft clipping, and H is hard clipping. For double-ended reads, the CIRI algorithm considers a pair of reads, one of which can be mapped to circRNA, and the other needs to be mapped to circRNA. For single exon ring formation, or the ring structure formed by “long exon 1-short exon-long exon 2”, the CIGAR value should be xS/HyMzS/H and (x + y) S/HzM or xM (y + z) S/H; CIRI software can separate these two situations very well. For splicing signals (GT, AG), CIRI will also consider other weak splicing information, such as AT-AC. The algorithm will extract exon boundary positions from GTF/GFF files and use known boundaries to filter false positives.

Currently, the main method for the identification of circRNAs involves sequence separation, alignment, and identification of GT-AG signals next to nodes. The software program Find circ [28] is a rapid and effective tool for identifying circRNAs. Since circRNA-forming splice sites cannot be directly compared to the genome, find_circ first establishes 20 bp anchor points at both ends of the reads in the genomic alignment. The anchor points are aligned as independent reads to the genome, and the program then searches for unique matching sites. If the alignment positions of the two anchor points are reversed in the linear direction, the reading of the anchor points is extended until circRNA binding is detected. If the sequences on both sides are GT/AG splicing signals, the region is judged to be a circRNA.

### 2.6. Analysis of the Differentially Expressed circRNAs

We performed statistics on the distribution of the length of circRNA in each sample and compared the position of circRNA and its source gene in the reference genome. The expression level of circRNA in each sample was calculated, and junction reads were used as the expression level of circRNA. The TPM method was used for standardization. The formula is as follows:TPM = Rj × 10^6^/([∑ki = 1Ri/Li] × Lj)
where R refers to junction reads of circular RNA, L: length of reads, and k: number of circular RNAs.

Gene expression is temporally and spatially specific. Depending on the relative levels of expression between the two groups, the DEGs were classified as upregulated genes or downregulated genes. The levels of expression of upregulated genes were higher in group B than in group A; conversely, those of downregulated genes were higher in group A than in group B. The degrees of up- and downregulation were relative and were determined by magnitudes of the differences between A and B.

### 2.7. Screening of Differentially Expressed circRNAs

When detecting differential expression of circRNAs, it is necessary to select appropriate differential expression analysis software according to the actual situation. DESeq [36] is suitable for experiments with biological repetitions. It can perform differential expression analysis between sample groups to obtain the differentially expressed circRNA set between two biological conditions; therefore, we used differentially expressed DESeq for differential expression analysis. In the process of detecting differential expression of circRNA, the fold change is greater than or equal to 2 and *p* value is less than 0.05 as the screening criteria [37] (Appendix A). The fold change indicates the ratio of the expression levels between the two samples (groups). Since differential expression analysis of circRNAs is an independent hypothesis test for a large amount of circRNA expression data, there is a risk of false positives. Therefore, in the analysis process, a significant *p* value was used as a key indicator for differential circRNA expression screening. The set of genes revealed by the differential expression analysis was called the differentially expressed gene (DEG) set.

### 2.8. CircRNA Source Gene Enrichment Analysis

The R package clusterProfiler [38] was used to analyze the biological process, molecular function, and cell component enrichment of genes from different circRNA sources. ClusterProfiler analyzes whether the differentially expressed circRNA-derived genes have occurred in a certain pathway, which is the enrichment analysis of the KEGG pathway for differential expression of circRNA. We tested the statistical enrichment of the DEGs in KEGG [39] pathways using KOBAS [40] software. GO terms and KEGG pathways with *p* values < 0.05 were considered to be significantly enriched.

### 2.9. RNA Extraction, Primer Design and RT-qPCR Detection

Total RNA was extracted using TRIzol reagent (Tiangen, Beijing, China) and an miRcute mRNA Extraction Kit according to the manufacturer’s recommended protocol. A NanoDrop ND-2000 spectrophotometer (NanoDrop Technologies, Wilmington, DE, USA) was used to measure the concentration of the RNA. Total RNA was then converted to cDNA using a FastQuant RT kit (containing gDNase; Tiangen, China) according to the manufacturer’s instructions. The primers for mRNA (FSHb and LHb) were designed by NCBI Primer-BLAST and verified by Primer 3 (https://bioinfo.ut.ee/primer3-0.4.0/, accessed on 20 July 2019) (Appendix A). The primers for five circRNAs were designed by RiboBio Biotech Co., Ltd. (Guangzhou, China). Due to the limitations of next-generation sequencing, RT-qPCR was then performed using SuperReal PreMix Plus (SYBR Green; Tiangen, China) and a Mastercycler ep Realplex 2 system (Eppendorf, Hamburg, Germany) according to the manufacturers’ instructions.

### 2.10. CircRNA and miRNA Target Prediction

CircRNAs have miRNA binding sites, and can be used as miRNA molecular sponges to regulate target genes. Previously, our team discovered many miRNAs that have a targeting relationship with FSHb [34]. To explore whether circRNAs can regulate the expression of FSH or LH through these miRNAs, we used miRanda software to predict the targeting relationship between circRNAs and miRNAs. miRanda is often used to predict miRNA target genes [41,42]. The input files are the differentially expressed circRNA sequence and the miRNA sequence found in our previous research that can target FSHb.

### 2.11. Statistical Analysis

RT-qPCR and ELISA data were analyzed using SPSS 22.0 software. The significance of differences was determined by one-way ANOVA, where *p* < 0.05 was considered to indicate significance.

## 3. Results

### 3.1. Changes in Hormone and Gene Expression Levels in the Pituitary Gland after GnRH Treatment

To verify whether injection of GnRH affected the Fshβ and Lhβ genes in rats and to further understand the regulatory effects of GnRH on reproduction, we determined the expression levels of Fshβ and Lhβ in the anterior pituitaries of rats by RT-qPCR. The serum of the rats was assayed by ELISA to determine the levels of FSH and LH.

We divided the remaining six rats into a control group and an experimental group. The control group was injected subcutaneously with 0.2 mL of physiological saline, and the experimental group was injected with 0.2 mL of GnRH solution (1 μg/mL) prepared from Gonadorelin for injection (San Sheng, Ningbo, China). The rats were given food ad libitum. Two hours later, RNA was extracted from pituitary tissue, blood was collected from the tail veins, and the expression levels of Fshβ and Lhβ in the two groups of rats were measured by fluorescence analysis. It was found that injection of the GnRH solution increased Fshβ expression levels by 1.44-fold (*p* < 0.05, Figure 1A) and Lhβ expression levels by 2.13-fold (*p* < 0.05, Figure 1B). We then measured FSH and LH secretion at the protein level. Interestingly, the secretion of FSH and LH was consistent with the expression of Fshβ and Lhβ in the GnRH group compared to the control group. Compared with the control group, the GnRH group had a two-fold increase in the level of FSH (Figure 1C) and a four-fold increase in the level of LH (Figure 1D).

### 3.2. RNA Sequencing of circRNAs in GnRH-Treated Rats

Samples were collected after GnRH treatment, circRNAs were identified, and finally, the sources and functions of the differentially expressed circRNAs were analyzed. The specific protocols are listed in the materials and methods, and a flow chart of our experiment is shown in Figure 2A. To visualize the relative mRNA and circRNA expression levels in the pituitary gland, we constructed a pie chart of all the RNA types (Figure 2B). At the same time, we analyzed the sources of the circRNAs and found that 80% came from exons, 8% from introns, and 12% from intergenic regions (Figure 2C) (Appendix A). To examine changes in circRNA expression between the GnRH-treated and control groups, we performed a statistical analysis (Figure 2D) (Appendix A). The results showed that 539 circRNAs were expressed only in the GnRH-treated group, while 633 circRNAs were expressed only in the control group. Most circRNAs were 400~1400+ nt in length; these circRNAs were mainly exonic circRNAs. CircRNAs with lengths ranging from 1400–2200 nt were also mainly derived from exons. The 2200~3000 nt length range included the fewest circRNAs. In addition, circRNAs longer than 3000 nt were mainly derived from intergenic regions (Figure 2E) (Appendix A). Finally, we labeled the detected circRNAs and mRNAs in the genome and constructed a Circos plot (Figure 2F) (Appendix A).

### 3.3. Enrichment of Differentially Expressed circRNAs

We used DEseq2 for differential expression clustering analysis of four sets of circRNAs. In comparisons between the GnRH-treated group and the control group, a fold change greater than or equal to 2 and a *p* value less than 0.05 were used as screening criteria for significantly differentially expressed circRNAs, and the results are plotted in Figure 3A (Appendix A). Ultimately, 14 differentially expressed circRNAs were obtained; among these, eight circRNAs were significantly upregulated, and six circRNAs were significantly downregulated, as shown in the volcano plot (Figure 3B) (Appendix A).

We then used GO and KEGG analyses to examine the enrichment of all the circRNAs and the differentially expressed circRNAs. GO enrichment analysis was first performed on the differentially expressed circRNAs and all the other circRNA source genes (Figure 3C). A directed acyclic graph of the top GO terms was constructed. Ultimately, seven circRNAs were annotated with GO terms including GO: 0043226, GO: 0044464, GO: 0065007, GO: 0016020, and others (Appendix A). Finally, we performed KEGG pathway enrichment analysis of the differentially expressed circRNAs (Figure 3D) (Appendix A). The differentially expressed circRNAs were enriched in one KEGG pathway, pancreatic secretion (ko04972).

### 3.4. Validation of Highly Expressed circRNAs

To verify the accuracy of the circRNA-seq results, we randomly selected eight highly expressed circRNAs and then detected the relative expression levels of these circRNAs in the pituitary in the GnRH-treated group and the control group by RT-qPCR. A total of three upregulated circRNAs and two downregulated circRNAs were identified in the GnRH-treated group compared to the control group. The final quantitative results were highly consistent with the RNA-seq results (Figure 4). These circRNAs are named according to their location on the chromosome.

### 3.5. Interactions between circRNAs and miRNAs

CircRNAs contain multiple miRNA-binding sites, and in the early stages of research in our laboratory, numerous miRNAs that may have target relationships with Fshβ [34]. It has also been determined that rno-miR-186-5p, rno-miR-21-3p, rno-miR-433-3p, and rno-miR-7a-5p indeed target Fshβ [34,43,44]. We thus used miRanda software to predict the differentially expressed circRNAs that might target these miRNAs and to visualize them based on their interactions (Figure 5) (Appendix A).

## 4. Discussion

Increases in GnRH cause the release of LH and FSH; similarly, the concentrations of LH in female sparrows increase after injection of GnRH [45]. Furthermore, it has been experimentally verified that suspensions of GnRH analogs are better able to promote FSH and LH secretion than solutions of GnRH itself [46]. The hypothalamus is capable of secreting two types of GnRH, GNRH1 and GNRH2, both of which can increase the expression levels of Fshβ and Lhβ, although GNRH2 may require the GNRH1 receptor to exert its effects [47]. In cattle, Kadokawa H et al. found that gonadotropin-releasing hormone receptor (GNRHR) aggregates on the surfaces of gonadotropic cells and that GnRH promotes LH and FSH secretion, wherein GNRHR concentrations increase with increasing concentrations of GnRH [48]. Debra M. Yeh et al. used mice to demonstrate that both GNRH1 and pituitary adenylate cyclase-activating polypeptide (PACAP) can promote the secretion of FSH and LH through this pathway [49]. Wells R et al. found that treatment with GnRH agonists promoted the secretion of FSH and LH in the pituitaries of pigs and affected the growth of testes in boars [50]. In our study, changes in FSH and LH in rats after GnRH treatment were detected by ELISA. The results revealed that the expression levels of Fshβ and Lhβ in the pituitary were increased after GnRH treatment; Fshβ was increased by 1.44-fold, and Lhβ was increased by 2.13-fold. These findings are consistent with the results of other experiments.

GnRH is secreted by the hypothalamus and plays a central role in the hypothalamic-pituitary-gonadal axis [51]. GnRH also regulates the reproductive capacity of animals [52]. In addition, GnRH and GNRHR have a confirmed relationship with cancer, providing the possibility of cancer treatment through targeting of these molecules [51,53]. GnRH injection can impair the formation of myocardial blood vessels in patients with heart disease, deepen the state of hypoxia in patients with cardiomyopathy, and elicit antiangiogenic effects in tumors [54]. Continuous use of GnRH can alleviate winter anovulation in mares and can be used in animal reproduction applications [55]. Furthermore, injection of GnRH analogs in mice may affect endometrial homeobox (HOX) a10 DNA methylation and endometrial receptivity [56].

Sequence analysis of the rat Lhβ basic promoter revealed a TGACCTTGT sequence from nucleotides -127 to -119. Orphan nuclear receptor steroidogenic factor-1 (SF-1) binds to and activates the Lhβ promoter and has been found to participate in the expression of Lhβ [57]. Hypothalamic GnRH also regulates the expression of the Lhβ; furthermore, there are two putative binding sites for SP1 (a trizinc finger transcription factor) in one region of Lhβ, and the binding of SP1 at these sites plays a role in the GnRH reactivity of Lhβ [58]. During the process by which GnRH promotes LH release, early growth response 1 (EGR1) and EGR2 stably induce activation of the Lhβ promoter at high GnRH pulse frequencies. In contrast, at low pulse frequencies, NGFI-A binding protein members inhibit Egr-induced activation of LHb [59]. GnRH1 rapidly stimulates various MAPK cascades, and the ERK1/2 and p38 pathways induce Fshβ gene expression. In addition, it has been found that the GNRH1 and activin A pathways cluster at the level of the high-affinity GNRH1-stimulating activator (AP-1) site. This pathway of GNRH1 and activin A also regulates FSHb expression [60]. cAMP response element-binding protein (CREB) is also involved in the regulation of Fshβ gene expression. During the process by which GnRH is repressed to regulate CREB, the protein kinase A (PKA)-mediated signaling pathway mediates GnRH activation of CREB at low GnRH pulse frequencies [61]. However, whether the Fshβ and Lhβ genes are regulated by circRNAs has not been reported. In this study, a total of 14 differentially expressed circRNAs were identified in the group treated with GnRH. These differentially expressed circRNAs may act on the Fshβ and Lhβ genes.

Xiao Yang [62] used next-generation sequencing (NGS) analysis to sequence circRNAs on an Illumina HiSeq 2500 instrument to identify circRNAs associated with bladder cancer. Kuei-Yang Hsiao [63] also sequenced RNA in a study on colon cancer; rRNA was removed using RiboMinusTM technology (Invitrogen), and the remaining RNA was sequenced using a SOLiDTM sequencer system. Xiaofeng Song [64] used an Illumina HiSeq 2000 for circRNA-seq of gliomas and identified circRNAs using the UROBORUS tool. Notably, traditional sequencing techniques cannot distinguish between circRNAs and linear RNAs [65]. Many algorithms are currently available that predict and identify circRNAs and thus facilitate genome-wide characterization of circRNAs. For example, Xintian You et al. developed the software package acfs. Acfs enables fast and accurate identification of circRNA, can be widely applied, and has a particularly low error rate [66]. Metge F et al. proposed that circRNAs can be effectively identified and that candidate circRNA structures can be fully characterized with the FUll circular RNA CHaracterization from RNA-seq (FUCHS) pipeline [67]. Furthermore, Gao Y et al. created a detection tool, CIRI2, that has very balanced sensitivity, reliability, duration, and RAM [68]. We verified our circRNA-seq results by RT-qPCR, and the quantitative results were consistent with the sequencing results.

In our study, we used RNA sequencing to sequence RNA. rRNA was removed using a Ribo-Zero rRNA Removal Kit (Epicenter, Madison, WI, USA). The sequencing library was prepared with an Illumina^®^ NEBNext^®^ UltraTM Directed RNA Library Preparation Kit (NEB, USA). Finally, sequencing was performed on the Illumina platform.

CircRNAs are a new class of long noncoding RNAs characterized by a covalent bond that connects the 3′ and 5′ ends to form a closed-loop structure that prevents exonucleolytic degradation by RNase R [69,70]. Rybak-Wolf A et al. showed that circRNAs are highly expressed in the mammalian brain and that they play important roles [71]. In this study, we identified 14 differentially expressed circRNAs in GnRH-treated rats that may have specific functions in the reproductive development of animals. Cheng J et al. similarly found that circRNA-103827 and circRNA-104816 may be involved in glucose metabolism, the mitotic cell cycle, and ovarian steroidogenesis and proposed that these circRNAs can be used to improve female infertility management [72]. Veno MT et al. reported the existence of differentially expressed circRNAs during pig embryonic brain development, suggesting that circRNAs may affect this process [73]. In addition, another study incorporating GO analysis of circRNAs indicated that circRNAs may be involved in spermatogenesis, sperm motility, and fertilization [72]. Therefore, we hypothesize that differentially expressed circRNAs are important regulators of animal reproductive development. CircRNAs also act as molecular sponges of miRNAs to regulate cancer. The circRNA-010567-miR141-TGF-β1 axis may play an important role in mouse myocardial fibrosis [74]. In addition, circRNA-100782 may regulate pancreatic cancer cell proliferation through interleukin 6 (IL-6) and signal transducer and activator of transcription 3 (STAT3) as a molecular sponge of microRNA-124 [75]. Another study suggested that circRNA-100876 is differentially expressed in both tumor and nontumor tissues of patients with non-small-cell lung cancer (NSCLC), suggesting that circRNA-100876 may be a therapeutic target for NSCLC [76].

GO enrichment analysis indicated that the differentially expressed circRNAs were mainly associated with reproductive processes, responses to stimuli, and protein-binding transcription factor activity. Related reports have indicated that GnRH and GNRHR play important roles in pathological processes in the female reproductive system [77]. In addition, studies have shown that under GnRH stimulation, the pituitary releases FSH, which stimulates the secretion of testosterone (T) [78]. Further studies have revealed that the RNA-binding protein ELAVL1 is one of the most abundant mRNA-binding proteins in cells and is involved in GnRH receptor-mediated regulation of LH secretion [79]. Thus, circRNA host genes may play key roles in the regulation of rat pituitary secretion after GnRH treatment. Our results may enrich the understanding of circRNA-mediated regulation of pituitary hormone secretion in rats.

## 5. Conclusions

In the present study, we identified a total of 1433 circRNAs in the anterior pituitaries of rats, and 14 circRNAs were differentially expressed in the GnRH-treated group compared with the control group. We also performed enrichment analysis of the differentially expressed circRNAs. In summary, we have provided a catalog of circRNAs in the rat pituitary and identified circRNAs that are differentially expressed in the pituitary between control and GnRH-treated rats. In addition, we predicted targeted relationships between differentially expressed circRNAs and Fshβ/Lhβ-associated miRNAs. Our results suggest that circRNAs in the anterior pituitaries of rats may have important biological effects and may be important regulators of animal reproductive development. This study provides important insights to enhance the understanding of circRNA-mediated regulation of secretion in the pituitary gland.

## Figures and Tables

**Figure 1 animals-11-02557-f001:**
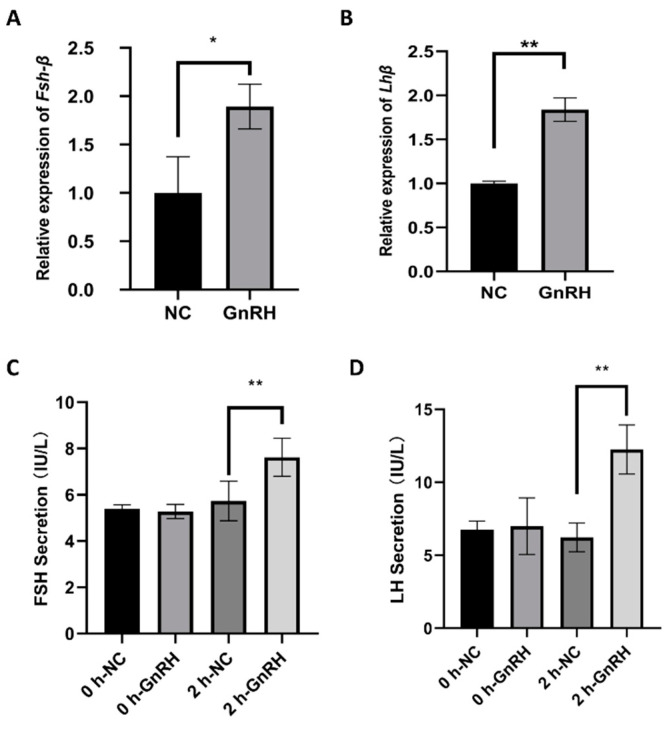
GnRH treatment upregulates the expression of Fshβ and Lhβ. After rats were treated with GnRH, Fshβ (**A**) and Lhβ (**B**) expression in the anterior pituitary was detected by RT-qPCR, and FSH and LH secretion was detected by ELISA (**C**,**D**). The data are shown as the mean ± SD of at least three independent experiments. Statistical significance was analyzed by one-way ANOVA (* *p* < 0.05; ** *p* < 0.01).

**Figure 2 animals-11-02557-f002:**
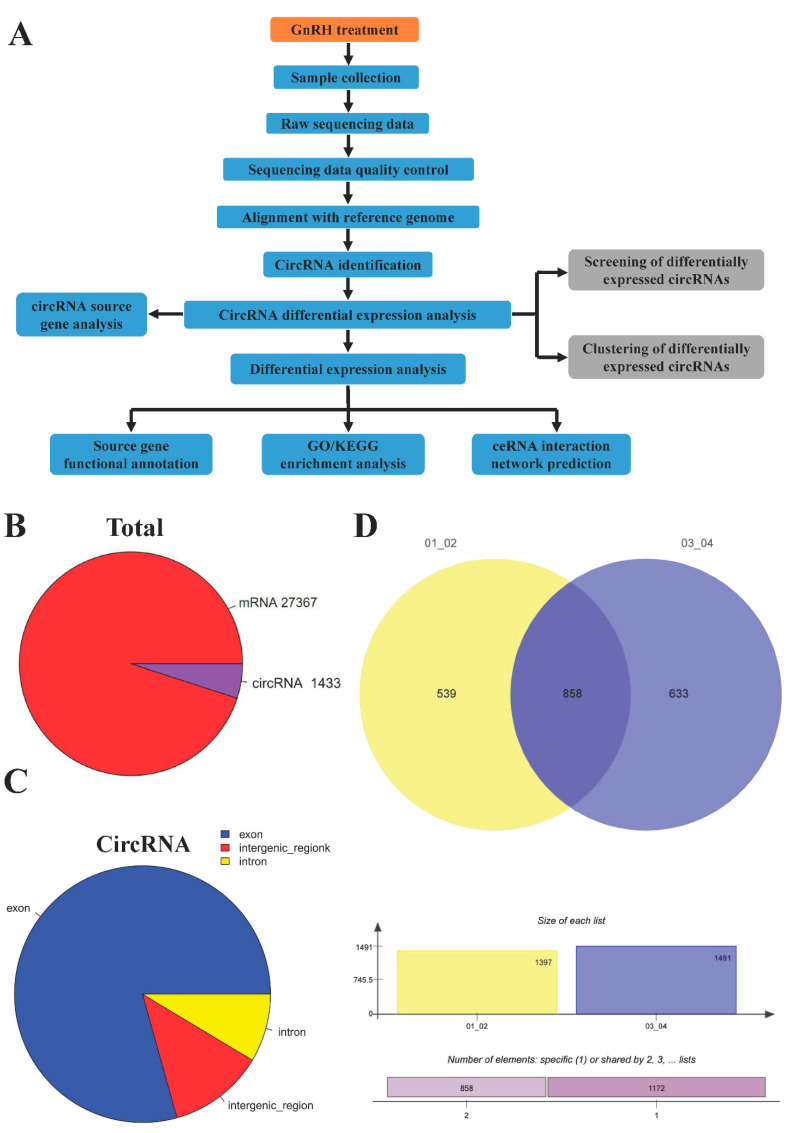
Deep sequencing of circRNAs in GnRH-treated rats. (**A**) Experimental flow chart. (**B**) Pie chart of mRNA and circRNA. (**C**) Statistical analysis of circRNA sources. (**D**) Statistical analysis of circRNA expression in the different groups. (**E**) Statistical analysis of circRNA length. (**F**) Circos plot containing circRNA and mRNA data.

**Figure 3 animals-11-02557-f003:**
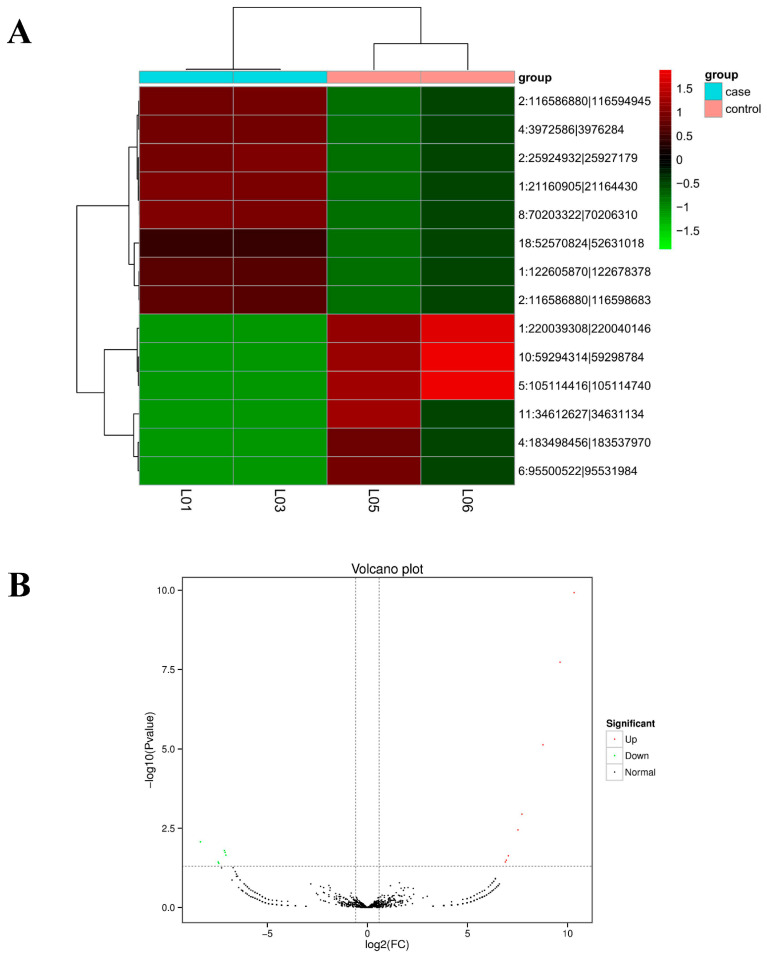
Enrichment analysis of the differentially expressed circRNAs. (**A**) Differential expression analysis of four groups of circRNAs. (**B**) CircRNA expression volcano plot. (**C**) GO analysis of the differentially expressed circRNAs. (**D**) KEGG pathway enrichment analysis of the differentially expressed circRNAs.

**Figure 4 animals-11-02557-f004:**
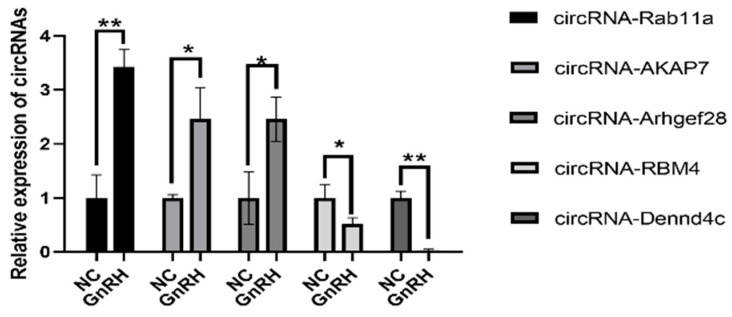
Validation of highly expressed circRNAs. Highly expressed circRNAs were detected by RT-qPCR. The data are shown as the mean ± SD of at least three independent experiments. The data are shown as the mean ± SD of at least three independent experiments. Statistical significance was analyzed by one-way ANOVA (* *p* < 0.05; ** *p* < 0.01).

**Figure 5 animals-11-02557-f005:**
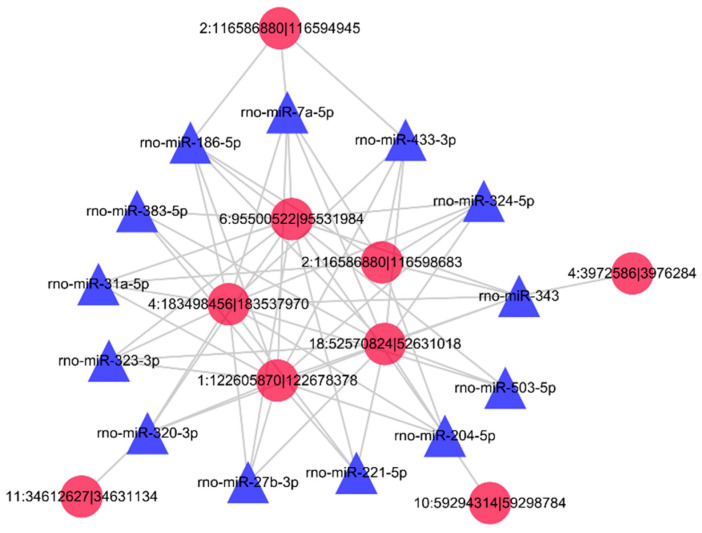
Interactions between circRNAs and miRNAs. The network was created with the miRanda program. The relationships between differentially expressed circRNAs and miRNAs that may target FSHb and LHb were predicted and are represented in the network. The red circles in the figure represent the differentially expressed circRNAs, and the blue boxes represent the miRNAs that may target Fshβ and Lhβ.

## Data Availability

The data contained in the article has been submitted to the Appendix A.

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
