# Peer review of "Identification of Circular RNAs in the Anterior Pituitary in Rats Treated with GnRH"

_animals, 2021, doi:10.3390/ani11092557_

Round 1
Reviewer 1 Report
Dear authors,
L110-111: you say that you have 12 rats, and you divide them in two groups: treated and controls? How can the two groups be of three rats each? (12:2=6)
L129-132: I don't completely agree with you when you say that RNAseq gives false positive results in the presence of splicing: i) first of all, is splicing taking place also with circRNA? (for instance, this is generally not an issue with miRNA, which are only ~22 bps long); ii) there are bioinformatics methods and software capable of accounting for splicing (e.g. exon/intron junctions) when assembling and mapping RNA/cDNA reads to produce count tables. Maybe you ought to describe this better in the your methods, at the RNA sequencing section; iii) RT-qPCR may be used to validate some of the results, as you say, and I would keep it at that: you used to molecular methods to confirm the presence of circRNA in the pituitary gland of rats
L142-147: I don't think that samples "needed to be pooled", you could have sequenced the individual samples obtaining 12 sets of RNA count data, instead of pooling their cDNA and end up with only 4 sets of sequencing data. In any case, if this is what you did, please describe it carefully and precisely, together with the limitations linked to pooling (yo reduce artificially the phenotypic variability, but also reduce the variability in gene/circRNA expression). Additionally, you could have pool your data afterwards, statistically, while retaining individual sequence data to allow for comparison between pooled vs non-pooled analysis.
L180-181: you could publish FDR values alongside your p-values, without changing the filtering criteria, but providing more information to the reader
L194-195: please explain in greater detail what you did here, why you used miRanda, why you produced miRNAs, and what is the relationship between miRNAs and circRNAs in the context of this article (future articles are not relevant here)
Author Response
1、L110-111: you say that you have 12 rats, and you divide them in two groups: treated and controls? How can the two groups be of three rats each? (12:2=6)
Thank you for your comment. We did not describe it clearly, and we have modified it in the original text.
L 111-117:
We randomly divided the rats into four groups, and each group contained three rats. Two of them were the experimental group (L01, L02), and the other two were the control group (L03, L04). The control group was injected subcutaneously with 0.2 ml of normal saline. According to the European Medicines Agency's summary report on GnRH, the experimental group was injected with 0.2 ml of prepared GnRH solution (containing 0.2 µg of drug), which was prepared from Gonadorelin for injection (San Sheng, Ningbo, China).
2、L129-132: I don't completely agree with you when you say that RNAseq gives false positive results in the presence of splicing: i) first of all, is splicing taking place also with circRNA? (for instance, this is generally not an issue with miRNA, which are only ~22 bps long); ii) there are bioinformatics methods and software capable of accounting for splicing (e.g. exon/intron junctions) when assembling and mapping RNA/cDNA reads to produce count tables. Maybe you ought to describe this better in the your methods, at the RNA sequencing section; iii) RT-qPCR may be used to validate some of the results, as you say, and I would keep it at that: you used to molecular methods to confirm the presence of circRNA in the pituitary gland of rats
Thank you for your correction. We have corrected the description order in the materials and methods. There is splicing during the formation of circRNA; in the process of circRNA identification, CIRI is usually used to identify circRNA and reduce false positive results. We have modified the original text; later, to determine the presence of circRNA in the rat adenohypophysis and the accuracy of the sequencing results, we used RT-qPCR for verification.
L 137-149:
2.5 CircRNA identification
The CIRI [35] software compares with the reference gene sequence to generate a SAM file, and analyzes the CIGAR value in the SAM file, scanning PCC signals (paired chiastic clipping signals) from the SAM file. The characteristic CIGAR value in the junction reads is xS/HyM or xMyS/H, where x and y represent the number of bases, M is mapping, S is soft clipping, and H is hard clipping. For double-ended reads, the CIRI algorithm considers a pair of reads, one of which can be mapped to circRNA, and the other needs to be mapped to circRNA. For single exon ring formation, or the ring structure formed by "long exon 1-short exon-long exon 2", the CIGAR value should be xS/HyMzS/H and (x+y) S/HzM or xM(y+z) S/H, CIRI software can separate these two situations very well. For splicing signals (GT, AG), CIRI will also consider other weak splicing information, such as AT-AC. The algorithm will extract exon boundary posi-tions from GTF/GFF files and use known boundaries to filter false positives.
3、L142-147: I don't think that samples "needed to be pooled", you could have sequenced the individual samples obtaining 12 sets of RNA count data, instead of pooling their cDNA and end up with only 4 sets of sequencing data. In any case, if this is what you did, please describe it carefully and precisely, together with the limitations linked to pooling (yo reduce artificially the phenotypic variability, but also reduce the variability in gene/circRNA expression). Additionally, you could have pool your data afterwards, statistically, while retaining individual sequence data to allow for comparison between pooled vs non-pooled analysis.
Thank you for your opinion. Indeed, performing separate sequencing of 12 samples, followed by nonmerged and combined analysis will significantly reduce individual differences, but due to financial constraints, we cannot complete the sequencing of 12 samples. Therefore, we adopted a compromise method, which combines the samples of each group before sequencing, and finally obtained four groups of relatively stable samples for subsequent sequencing analysis. The RT-qPCR results can also illustrate the accuracy of our sequencing results.
4、L180-181: you could publish FDR values alongside your p-values, without changing the filtering criteria, but providing more information to the reader
Thank you for your correction. We have published the FDR value and P value together and added it to the supplementary file.
5、L194-195: please explain in greater detail what you did here, why you used miRanda, why you produced miRNAs, and what is the relationship between miRNAs and circRNAs in the context of this article (future articles are not relevant here)
Thank you for your correction. We have modified the original text.
L 207-214
2.10 CircRNA and miRNA target prediction
CircRNAs have miRNA binding sites, and can be used as miRNA molecular sponges to regulate target genes. Previously, our team discovered many miRNAs that have a targeting relationship with FSHb [34]. To explore whether circRNAs can regu-late the expression of FSH or LH through these miRNAs, we used miRanda software to predict the targeting relationship between circRNAs and miRNAs. miRanda is often used to predict miRNA target genes [41, 42]. The input files are the differentially ex-pressed circRNA sequence and the miRNA sequence found in our previous research that can target FSHb.
L 301-345
CircRNAs contain multiple miRNA-binding sites, and in the early stages of re-search in our laboratory, numerous miRNAs that may have target relationships with Fshβ [34]. It has also been determined that rno-miR-186-5p, rno-miR-21-3p, rno-miR-433-3p and rno-miR-7a-5p indeed target Fshβ [34, 43, 44].
Refrences:
- Han DX, Sun XL, Xu MQ, Chen CZ, Jiang H, Gao Y, et al. Roles of differential expression of microRNA-21-3p and microRNA-433 in FSH regulation in rat anterior pituitary cells. Oncotarget. 2017;8(22):36553-65.
- Wang CJ, Guo HX, Han DX, Yu ZW, Zheng Y, Jiang H, et al. Pituitary tissue-specific miR-7a-5p regulates FSH expression in rat anterior adenohypophyseal cells. PeerJ. 2019;7:e6458.
- Han DX, Xiao Y, Wang CJ, Jiang H, Gao Y, Yuan B, et al. Regulation of FSH expression by differentially expressed miR-186-5p in rat anterior adenohypophyseal cells. PLoS One. 2018;13(3):e0194300.

Reviewer 2 Report
I prefer no review this paper again.
Author Response
Sorry to disturb you.
Reviewer 3 Report
Submitted for review this paper by Guo et al. deals with adenohypophyseal circular RNA expression in rats treated with GnRH. The study is well done, I suggest to implement the work with some bibliographic references. These are my suggestions:
L14 in which tissue are the analysis performed?
L41 add space before 2. L46 add space before brackets also at L49, 51, 52, 58, 61, 69, 71, 127, L178, 174
L193, 197 check the font type.
L49-51: GnRH regulates the production of luteal progesterone and testosterone in different species
Animals (Basel). 2021 Jan 25;11(2):296. doi: 10.3390/ani11020296.
Biol Reprod. 2012 Aug 23;87(2):45. doi: 10.1095/biolreprod.112.099598.
Domest Anim Endocrinol. 2011 Jan;40(1):51-9. doi: 10.1016/j.domaniend.2010.08.006.
Theriogenology. 2020 Aug;152:1-7. doi: 10.1016/j.theriogenology.2020.04.006.
L90 delete Han et al.
L177-184, why in bold?
L262 delete the point before the brackets.
Author Response
Submitted for review this paper by Guo et al. addresses adenohypophyseal circular RNA expression in rats treated with GnRH. The study is well done, I suggest to implement the work with some bibliographic references. These are my suggestions:
1、L14 in which tissue are the analysis performed?
Thank you for your correction. We have revised the original text.
L13-15:
In this study, we performed high-throughput sequencing of GnRH-treated rats to identify differentially expressed circRNAs in the anterior pituitary.
2、L41 add space before 2. L46 add space before brackets also at L49, 51, 52, 58, 61, 69, 71, 127, L178, 174
Thank you for your correction. This is our negligence. A space has been added before each bracket.
3、L193, 197 check the font type.
Thank you for your correction. We have modified the font type.
L208,L217:
2.10 CircRNA and miRNA target prediction
2.11 Statistical analysis
4、L49-51: GnRH regulates the production of luteal progesterone and testosterone in different species
Animals (Basel). 2021 Jan 25;11(2):296. doi: 10.3390/ani11020296.
Biol Reprod. 2012 Aug 23;87(2):45. doi: 10.1095/biolreprod.112.099598.
Domest Anim Endocrinol. 2011 Jan;40(1):51-9. doi: 10.1016/j.domaniend.2010.08.006.
Theriogenology. 2020 Aug; 152:1-7. doi: 10.1016/j.theriogenology.2020.04.006.
Thank you very much for your guidance. Here we refer to your suggestions and add a description of the GnRH function.
L 49-54:
GnRH can participate in the regulation of a variety of reproductive processes in mammals. For example, the addition of GnRH agonists during artificial insemination can induce ovulation and improve embryo survival [9, 10]. GnRH regulates the production of luteal progesterone and testosterone in different species [11-14]. In addition, GnRH agonists can also participate in the reproductive regulation of mammals after pregnancy [15].
5、L90 delete Han et al.
Thank you for your comments. We have deleted "Han et al" from the original text.
L 91-94
Finally, based on the results of our previous research [34] identifying many miRNAs that may have targeting relationships with Fshβ and Lhβ, we predicted the differentially expressed circRNAs that may target these miRNAs.
6、L177-184, why in bold?
Thank you for your correction. This is our negligence. The bold text has been modified from the original text.
L 179-186:
In the process of detecting differential expression of circRNA, fold change is greater than or equal to 2 and P value is less than 0.05 as the screening criteria [37]. The fold change indicated the ratio of the expression levels between the two samples (groups). Since differential expression analysis of circRNAs is an independent hypothesis test for a large amount of circRNA expression data, there is a risk of false positives. Therefore, in the analysis process, a significant P value was used as a key indicator for differential circRNA expression screening. The set of genes revealed by the differential expression analysis was called the differentially expressed gene (DEG) set.
7、L262 delete the point before the brackets.
Thank you for your correction. We have deleted the dot in front of the brackets.
L 281-282:
The differentially expressed circRNAs were enriched in 1 KEGG pathway, pancreatic secretion (ko04972).
This manuscript is a resubmission of an earlier submission. The following is a list of the peer review reports and author responses from that submission.
Round 1
Reviewer 1 Report
Dear authors,
I read your article "Identification of circular RNAs in the anterior pituitary in rats treated with GnRH", which I find well written and interesting. I have some concerns about the experimental design and description of methods, whcih you can find in the following specific comments. English is fine.
Introduction
-------------
L39: it is thought? I am not an expert in endocrinology, but I think that there is enough evidence to say that we know that GnRH is produced in the hypothalamus
L46-48: I think that gonadotropins analogs are also used in livestock reproduction / veterinary obstetrics
L57: why "unique"?
L72: I guess it's RNA sequencing
Materials and Methods
---------------------
L95-105: information on the number of samples (the size of the two experimental groups) is lacking: please provide this information in the text
L99: is this amount of GnRH sufficient to observe a physiological effect? How did you determine the 0.2 micrograms of GnRH?
L112: were the primers designed to target circular RNA or small non-coding RNA in general, or were they designed for all RNA molecules irrespective of type?
L113: it is not clear why you did also RT-qPCR, since you already have RNAseq to measure relative expression levels in the two groups
L129: please explain better the pooling of samples: does this mean that you sequenced only two pooled samples (controles and treated)? Why did you choose to pool samples rather than having individual sequencing results?
L134: please highlight somehow the name of the software and add a reference/link
L143: did you normalize circRNA expression data (counts) for library size and/or gene length before defferential expression analysis? This is usually a necessary step in RNAseq, but maybe it does not apply to circRNA?
L146-149: it is not clear how you controlled for false positives. This is usually done by calculating FDR (false discovery rate), or by applying a stricter threshold on p-values depending on the number of tests (i.e. n. of circRNA loci tested), or by some resampling approaches (e.g. permutation test), etc. What was your approach? Onlu looking at p-value < 0.05 (+ fold-change > 2) may not be enough
L165-166: so you used miRanda to find circRNA targets? Is this appropriate? Additionally, you mention here miRNAs and mRNAs, but in previous sections you talked only of circRNA (DESeq, GO, KEGG ...). It is not clear if you analysed only circRNA ro also other RNA types
L168-169: it is not clear which analyses were carried out with SPSS 22.0: all? Also DESeq? You previously said that GO and KEGG analyses were performed with R: these too are statistical analyses. Please clarify and describe this more clearly
Results
-------
L195: why do you call it "Deep sequencing" here? If deep has some specific meaning (e.g. coverage etc.), this needs to be described in Methods
Figure 2: quality is very low, and I can not read well the figure subplots (labels etc.)
Reviewer 2 Report
In the present study, the authors identify a total of 1433 circRNAs in the anterior pituitaries of rats and evaluate the different expression after GnRH. This study provides important insights to enhance understanding of the circRNA-mediated regulation of secretion in the pituitary gland.
The manuscript is interesting and clearly written and hence accepted with minor revision.
I suggest a single revision:
clarifies why the 4 hours for biological sampling are chosen.
Reviewer 3 Report
This manuscript must be re-written and the data re-analysed to reach the standards of a research publication.